# CRAFT: A Neuro-Symbolic Framework for Visual Functional Affordance Grounding

**Zhou Chen**                                    zzc0053@auburn.edu
**Joe Lin**                                      jzl0277@auburn.edu
**Sathyanarayanan N. Aakur**                     san0028@auburn.edu
*CSSE Department, Auburn University, AL, USA*

**Editors:** Leilani H. Gilpin, Eleonora Giunchiglia, Pascal Hitzler, and Emile van Krieken

## Abstract

We introduce CRAFT, a neuro-symbolic framework for interpretable affordance grounding, which identifies the objects in a scene that enable a given action (e.g., "cut"). CRAFT integrates structured commonsense priors from ConceptNet and language models with visual evidence from CLIP, using an energy-based reasoning loop to refine predictions iteratively. This process yields transparent, goal-driven decisions to ground symbolic and perceptual structures. Experiments in multi-object, label-free settings demonstrate that CRAFT enhances accuracy while improving interpretability, providing a step toward robust and trustworthy scene understanding.

## 1. Introduction

Autonomous agents must reason not just about what objects are, but what they enable — how they can serve a user's goal. For example, when asked to "give me something to cut with," a robot must recognize that a knife, scissors, or even broken glass afford the action "cut," regardless of their category or label. This functional perspective reflects a Gibsonian affordance (Gibson, 2000, 2014): the actionable possibilities an environment offers relative to an agent's capabilities. Grounding such affordances requires integrating commonsense knowledge (e.g., knives cut), structural reasoning (e.g., sharpness matters), and visual grounding. Purely learning-based models, such as vision-language models (VLMs) (Radford et al., 2021), often falter in these open-world settings, lacking the symbolic scaffolding necessary to generalize across ambiguous or unseen contexts.

**Prior work** on affordance grounding rely on supervised labels or handcrafted knowledge bases, which constrain generalization to open-world tasks (Qu et al., 2024; Sawatzky et al., 2020). With the rise of vision-language models Radford et al. (2021), work explores zero-shot affordance reasoning via image-text alignment (Cuttano et al., 2024). However, such models lack structured reasoning and struggle in ambiguous or cluttered scenes (Chen et al., 2024). To improve robustness, newer approaches use large language models to extract affordance attributes via prompting (Tang et al., 2023) or bypass object labels by grounding from verbs (Nguyen et al., 2020). Others enhance interpretation with external knowledge sources or perception-action APIs (Mavrogiannis et al., 2023). Despite progress, aligning functional semantics with visual context across diverse tasks remains a challenging task.

To address these limitations, we propose CRAFT (Compositional Reasoning for Affordance Focused Traces), outlined in Figure 1. This neuro-symbolic framework unifies symbolic priors with visual evidence to ground functional affordances in ambiguous, open-world

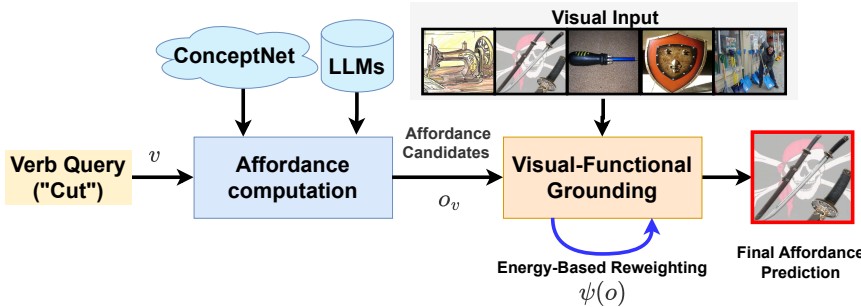

Figure 1: **CRAFT Overview.** Given a verb query $v$, symbolic priors generate affordance candidates $o_v$, which are grounded in visual input. Energy-based reweighting refines predictions for grounding functional affordances.

settings. Given a verb query and a set of unlabeled images, CRAFT retrieves candidate objects from external knowledge sources, such as ConceptNet (Liu and Singh, 2004) or large language models (Anil et al., 2023). It iteratively aligns them with image regions using visual-semantic similarity. These symbolic candidates are reweighted through a reasoning-guided energy function that captures both prior relevance and grounded support, producing interpretable and robust affordance predictions. Our **contributions** are three-fold: (i) we formalize functional affordance grounding as neuro-symbolic inference combining priors and visual alignment; (ii) We propose an iterative mechanism to refine symbolic hypotheses in multi-object, label-free settings; (iii) We introduce a rigorous benchmark with real-world images and affordance annotations, and evaluate against diverse baselines.

## 2. Our Approach

**Problem Formulation and Overview.** We study *functional affordance grounding*: given a verb query $v$ (e.g., "cut") and a set of candidate object images $\mathcal{I} = \{o_1, \ldots, o_n\}$, the goal is to identify objects that afford the action. Inspired by Gibsonian affordances, we introduce **CRAFT**, which frames this as an energy minimization problem, where each object $o$ is assigned an energy $E(v, o)$ indicating its functional compatibility with $v$; lower energy denotes better alignment. To estimate $E(v, o)$, we use ConceptNet to extract multi-hop reasoning paths from verbs to plausible object concepts (e.g., "cut" $\rightarrow$ "use" $\rightarrow$ "knife"), capturing commonsense affordances. These symbolic traces define a structured prior over plausible objects, which we integrate with neural visual grounding models (Radford et al., 2021). CRAFT combines these elements using iterative grounding to identify affordant objects.

**Affordance Graph Construction** To capture commonsense knowledge about functional affordances, we construct an affordance graph $\mathcal{G}=(\mathcal{V},\mathcal{E})$ from ConceptNet, where nodes $\mathcal{V}$ include both verbs (affordances) and object concepts, and edges $\mathcal{E}$ represent affordance-related relations (e.g., `UsedFor`, `CapableOf`). Given a verb query $v$, we extract a local subgraph $\mathcal{G}_v$ by traversing affordance-relevant paths of bounded depth and filtering for object-like leaf nodes using ConceptNet's node type metadata and part-of-speech heuristics. Each object node $o$ in $\mathcal{G}_v$ is scored by aggregating path-based evidence strength, which serves as a prior for grounding by providing object hypotheses and structured reasoning traces.

**Energy-Based Visual Grounding**   Given a verb query $v$ and a set of candidate object images $\{x_i\}_{i=1}^n$, our goal is to identify the image $x^*$ that best affords $v$. We formulate this as minimizing an energy function $E(v, x_i)$ that reflects how well $x_i$ functionally aligns with $v$. Each candidate image $x_i$ is encoded using a pretrained vision-language model (e.g., CLIP) into an embedding $f(x_i) \in \mathbb{R}^d$. Similarly, for each object label $o \in \mathcal{G}_v$, we obtain a text embedding $g(o)$ and define its affordance compatibility with $x_i$ as $s(o, x_i) = \cos(g(o), f(x_i))$. We define the grounding energy of $x_i$ with respect to verb $v$ as:

$$E(v, x_i) = -\max_{o \in \mathcal{G}_v} [\phi(o, v) \cdot s(o, x_i)]$$

where $\phi(o, v)$ is the reasoning-based prior score assigned to object $o$ for verb $v$ from the affordance graph. The object image with the lowest energy is selected: $x^* = \arg\min_{x_i} E(v, x_i)$. This formulation fuses structured priors (via $\phi$) with visual similarity (via $s$), enabling alignment between affordance-driven expectations and image evidence.

**Iterative Reasoning and Re-ranking**   While the initial grounding identifies the best-matching object $x^*$ based on current priors, many affordances require contextual refinement, especially when $\mathcal{G}_v$ contains noisy or semantically diffuse concepts. To address this, we adopt an iterative re-ranking strategy. At each iteration $t$, we maintain a ranked list of candidate object labels $\mathcal{G}_v^{(t)} \subseteq \mathcal{G}_v$ and recompute scores based on updated priors and grounding feedback. We define an attention-like update rule over $\mathcal{G}_v^{(t)}$ as $\phi^{(t+1)}(o, v) \propto \phi^{(t)}(o, v) \cdot \exp(\lambda \cdot s(o, x_t))$, where $x_t$ is the top-ranked object at iteration $t$ and $\lambda$ controls the influence of grounding feedback. This soft re-weighting mechanism promotes concepts that align with high-confidence images and downweights irrelevant ones. The updated $\phi^{(t+1)}$ is used to recompute $E(v, x_i)$ for all $x_i$, and the process repeats until convergence. The final ranking strikes a balance between prior knowledge and visual evidence. The appendix provides qualitative visualizations of reasoning traces and affordance ego-graphs.

**Implementation Details.**   We use a pre-trained CLIP ViT-B/32 model (Radford et al., 2021) for computing visual-semantic similarity. Each candidate image $x_i$ is embedded via CLIP's image encoder, and candidate verb-object pairs $(v, o)$ are embedded using the text encoder. All similarities are computed in the normalized CLIP embedding space using cosine similarity. For prior extraction, we use two types of sources: ConceptNet and large language models (LLMs). For ConceptNet, all candidates connected to a verb node are ranked by path-based confidence scores. We retain the top 25 candidates using a two-stage sort: first by edge weight, then by similarity to the query verb (using ConceptNet NumberBatch (Speer et al., 2017)). For LLM priors, we use the top 10 objects, as constrained in the prompt.

## 3. Experimental Evaluation

**Experimental Setup.**   We evaluate on the dataset from Nguyen et al. (Nguyen et al., 2020), which provides verb-object affordance labels across 216 ImageNet (Deng et al., 2009) categories and 50 verbs. Each test episode samples 5 images from the validation set, mixing affordant objects and distractors. We test two settings: single-label (one correct) and multi-label (two correct), over 100 randomized episodes per verb. Performance is measured using top-1 accuracy, MRR, and nDCG to assess both precision and ranking quality.

Table 1: **Functional Affordance Performance** under single-label (one affordant) and multi-label (two affordants) settings from five object candidates.

| Type | Model | Accuracy@1 (Single-label) | MRR (Multi-label) | nDCG (Multi-label) |
|---|---|---|---|---|
| Oracle | Object Aware | 100.00% | 69.90% | 76.80% |
| | Affordance Aware | 73.80% | 68.70% | 82.70% |
| Learning-based | Afford-CLIP | 38.0% | 54.8% | 56.1% |
| | ResNet-RNN | 60.20% | 65.7% | 76.5% |
| Prior-only | ALGO | 42.52% | 52.38% | 53.33% |
| | Gemini-2.0-Flash | 42.80% | 57.50% | 59.70% |
| | GPT-4o | 45.30% | 57.80% | 60.40% |
| NeSy Grounding | CRAFT | 44.62% | 58.20% | 61.30% |
| | CRAFT + Gemini | 43.39% | 56.50% | 59.30% |
| | CRAFT + GPT-4o | 46.43% | 60.80% | 67.00% |

**Baselines.** We compare our method, CRAFT (Commonsense Reasoning for Affordance-Centered Functional Targeting), against four baseline types: (1) Oracle-based (Object Oracle, Affordance Oracle), which use ground-truth associations or object labels; (2) Prior-only models (ALGO (Kundu et al., 2024), Gemini (Anil et al., 2023), GPT4o (Hurst et al., 2024)), which rely on language-based or symbolic functional affordance candidates; (3) Learning-based models (Afford-CLIP (Radford et al., 2021), ResNet-RNN (Nguyen et al., 2020)) which learn affordances using learned priors (Afford-CLip: web-scale, ResNet-RNN (Nguyen et al., 2020): supervised) from annotated data and (4) Iterative grounding variants (CRAFT, CRAFT+Gemini, CRAFT+GPT4o), which combine symbolic or LLM-derived priors with the proposed grounding pipeline.

**Performance on Single-affordant Setting.** In this setting, models are evaluated on their ability to identify the one image (among $n=5$) that functionally affords the queried verb. Table 1 shows top-1 accuracy across models. As expected, the Oracle baselines (Object-aware and Affordance-aware) perform best at 100.00% and 73.80%, confirming that CLIP can support functional grounding when given exact labels or affordance-aligned prompts. Among prior-driven baselines, GPT4o leads (45.30%), followed by Gemini (42.80%) and ALGO (42.46%). Fully supervised ResNet-RNN models perform very well (60.20%), while the web-scale-trained AffordCLIP lags at 38.0%. These differences reflect each prior's nature: GPT4o yields concise, visually aligned suggestions; Gemini is broader but semantically relevant; ConceptNet, though rich in associations, often includes noisy or symbolic terms (e.g., "file," "bee") that lack clear visual affordances. When combined with our neuro-symbolic grounding framework, CRAFT+GPT4o reaches 46.43%—the best non-oracle result—by refining priors with visual evidence. CRAFT+Gemini (43.39%) and CRAFT+ConceptNet (44.62%) also outperform their prior-only counterparts.

**Performance on Multi-affordant Setting** In this setting, each episode includes two affordant objects, and models are evaluated on how well both are ranked. Table 1 reports MRR (how early the first relevant object appears) and nDCG (overall ranking quality). The oracle baselines perform best, with the Affordance-aware oracle achieving the highest nDCG

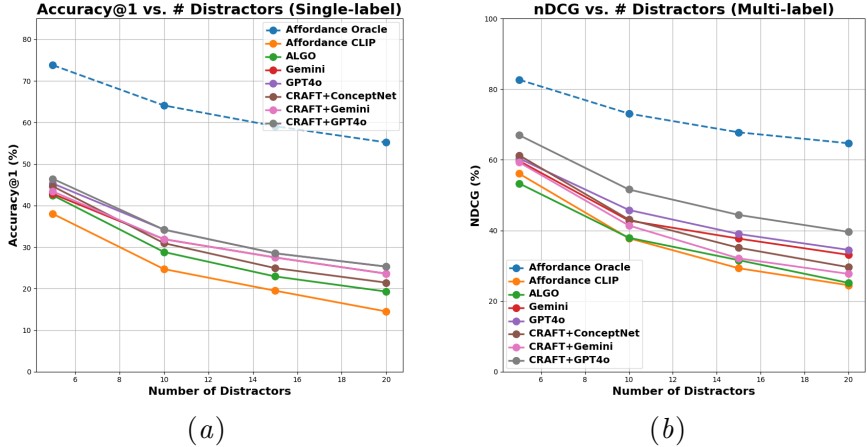

Figure 2: **Impact of distractors.** quantified as (a) Accuracy under the single-label setting. (b) nDCG under the multi-label setting, as a function of distractors.

(82.70%) and MRR (69.90%). Full supervision on in-domain semantics (ResNet-RNN) helps achieve close to oracle-level performance, while web-scale supervision (Afford-CLIP) provides a decent prior. Among prior-driven models, GPT4o (57.80%, 60.40%) and Gemini (57.50%, 59.70%) outperform ALGO, highlighting the strength of LLM-generated priors. CRAFT+GPT4o achieves the best non-oracle performance (60.80%, 67.00%), showing that iterative re-ranking improves both early precision and overall ranking. CRAFT+Gemini and CRAFT+ConceptNet also show consistent gains over their prior-only counterparts, confirming that integrating visual evidence enhances grounding even when priors are noisy.

**Effect of Distractors.** Figures 2(a) and (b) illustrate how grounding performance scales as the number of distractors increases from 5 to 20 candidates per episode. As expected, all models show declining accuracy (at different rates) and ranking quality. The oracle baselines exhibit the slowest decline, establishing clear upper bounds on achievable performance under perfect knowledge. Among prior-only models, GPT4o again consistently outperforms Gemini and ALGO, demonstrating greater robustness to distractor noise. Afford-CLIP, lacking explicit object priors, suffers a steep performance drop, highlighting its limited generalization. CRAFT-based models show clear improvements over their corresponding priors, even with noisier or semantically diffuse priors (ConceptNet, Gemini), highlighting CRAFT's core contribution: it amplifies strong priors and compensates for weaker ones, and helps handle uncertainty in open-world scenarios.

## 4. Conclusion, Limitations and Future Work

We presented CRAFT, a neuro-symbolic framework for grounding functional affordances by integrating structured commonsense priors with visual alignment via CLIP. CRAFT refines noisy object candidates through iterative energy-based reasoning, improving accuracy across symbolic and LLM-derived priors. Our findings underscore the value of interpretable reasoning in open-ended, label-free settings. While effective, CRAFT's reliance on external knowledge and multi-step inference introduces computational overhead. Future work will focus on distilling this process into lightweight surrogates for scalable affordance reasoning.

## Acknowledgments

This research was supported in part by the US National Science Foundation grants IIS 2348689 and IIS 2348690. The authors also thank Mr. Carson Bulgin for his inputs during the ideation stages and the anonymous reviewers for their constructive feedback.

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

## Appendix A

**Qualitative Interpretability.** Figure 3 visualizes selected reasoning traces used by CRAFT to ground affordance predictions. In examples (a–c), the paths connect the verb query (e.g., *cut*, *serve*) to semantically meaningful object concepts such as *tray* and *cleaver*, through relevant intermediate nodes like *serving* or *saw*, with interpretable edge relations (e.g., `UsedFor`, `HasSubevent`). These highlight how the model identifies affordant candidates via structured multi-hop reasoning. In contrast, examples (d–f) illustrate failure cases where the inferred paths traverse semantically loose or misleading associations (e.g., `Antonym`, `RelatedTo`) that connect to visually plausible but functionally incorrect objects like *notebook* or *cleaver* in irrelevant contexts. These traces underscore both the transparency and limitations of CRAFT's symbolic reasoning, allowing for post-hoc inspection and targeted debugging of erroneous predictions.

    **ConceptNet-based affordance graphs.** The ego-graphs in Figure 4 illustrate the symbolic neighborhood around each verb query—"cut," "eat," "play," and "write"—showing how ConceptNet captures affordance-relevant concepts through multi-hop relations. The verb is centered in yellow, with affordance candidates (as identified by the grounding system) highlighted in red, and intermediate concepts in blue. For compact verbs like "cut" and "eat," the graphs are small and semantically tight, with strong relational cues such as "UsedFor" and "RelatedTo" linking the verb to actionable concepts (e.g., "saw" or "cooking"). In contrast, "play" and "write" generate significantly denser graphs, reflecting their more diffuse semantics and broader conceptual scope. Despite this, many valid candidates like "violin" or "cricket" are correctly surfaced at the periphery. These graphs emphasize both the richness and noisiness of commonsense priors and showcase the traceable, interpretable nature of CRAFT's reasoning, offering a transparent lens into how abstract knowledge supports visual-functional inference.

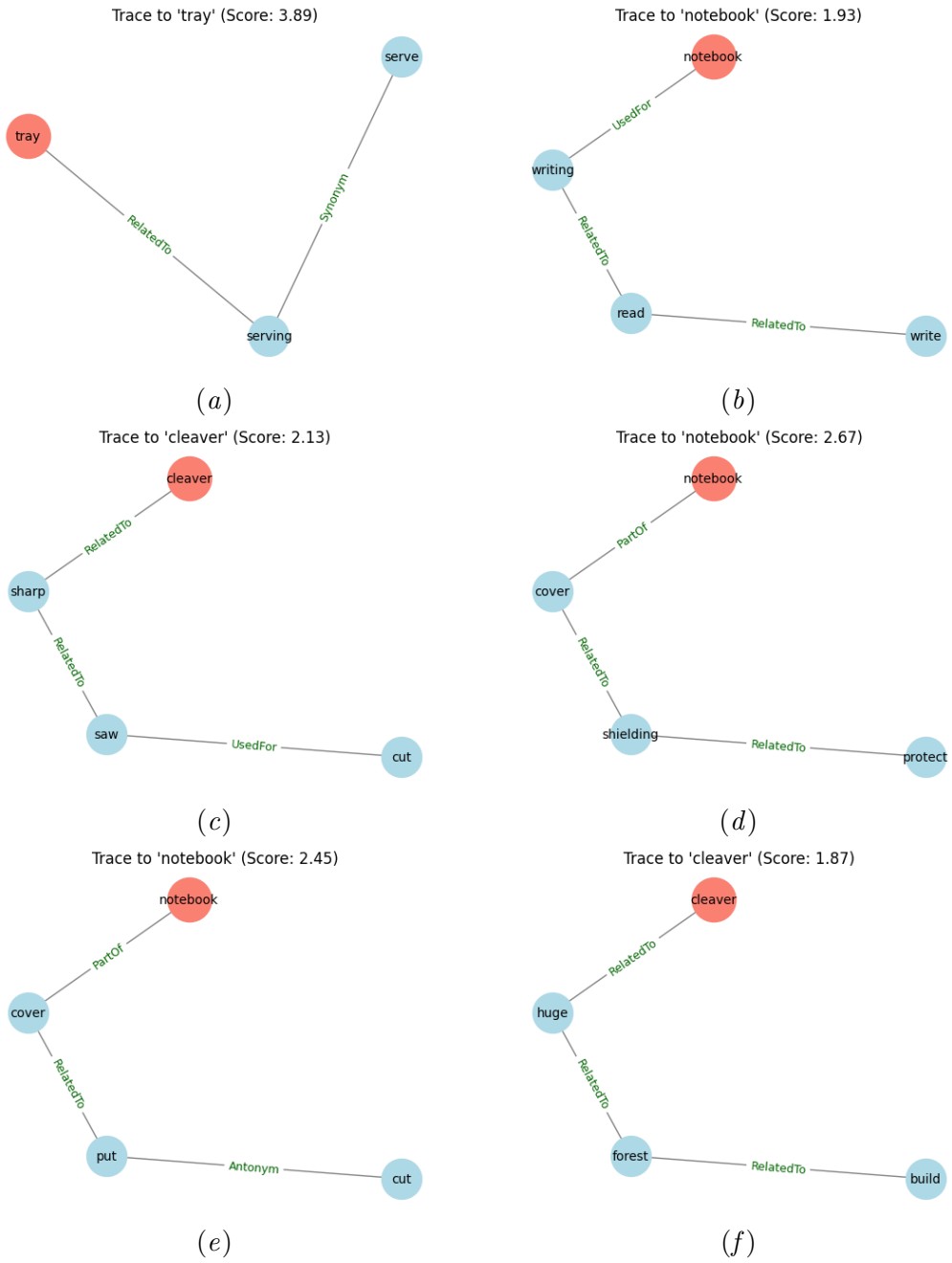

Figure 3: **Illustration of some reasoning traces that are correct (a-c) and irrelevant (d-f). Score refers to the normalized path weight in ConceptNet used to compute the reasoning-based prior score $\phi(o, v)$ .**

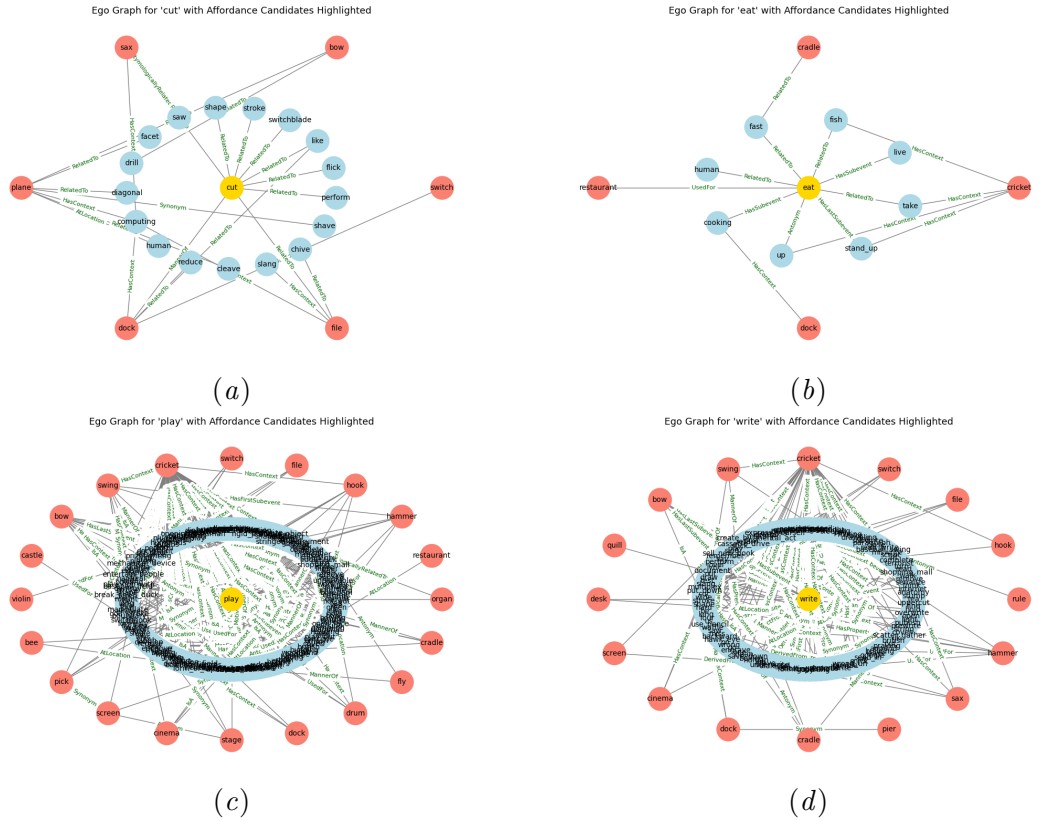

Figure 4: **Illustration of ego graphs used to generate affordance reasoning traces for verb query (a) "cut", (b) "eat", (c) "play", and (d) "write."**

