# OpenReview forum: "CRAFT: A Neuro-Symbolic Framework for Visual Functional Affordance Grounding"
_nesyconf.org/NeSy/2025/Conference_Phase_2 — NeSy 2025 - Phase 2 Poster_

### Official Review · Reviewer_jAAg · 2025-07-05
**review of Submission28**

**Rating:** 7
**Confidence:** 4

**Review:**

## summary of the paper

This paper introduces a neurosymbolic framework for doing affordance grounding, called CRAFT (Compositional Reasoning for Affordance Focused Traces). CRAFT uses a 3rd-party system called CLIP to do the visual aspect of the overall grounding task (finding images containing objects of certain classes). To map single-verb instructions (actions, or queries), such as "cut", to candidate object classes that afford that action, CRAFT uses either 1) a knowledge graph derived from ConceptNet that contains paths from verbs to object classes, or 2) LLMs (like GPT4). CRAFT calculates a score that measures the confidence of the visual grounding and a score that measures the confidence of the mapping from verb to candidate affordant object classes. These two scores are combined in a function that generates an overall score where lower is better. The authors refer to this as an energy function. They employ a scheme to iteratively refine these scores before making a final prediction. The image with the lowest overall (energy) score is selected as the predicted affordance object.

The performance of CRAFT is compared with several other systems on two related affordance grounding tasks.  Task 1 requires the system to pick the one image in five that contains an affordance object (one that affords the instruction action). Here performance is measured using an accuracy metric. Task 2 requires the system to pick the two images in five that contain an affordance object.
Here performance is measured using metrics MRR and nDCG.
CRAFT is shown to perform roughly the same as, and sometimes marginally better than, the other non-Oracle systems.


## evaluation of the paper

**clarity / use of language / quality of writing**

The paper is clear, nicely written and well organised. A pleasant read. Diagrams and text that are very helpful for understanding the research are relegated to an Appendix, however --- presumably to keep within the 5-page limit for short papers.

**novelty / originality**

I don't know the affordance grounding literature well enough to say for sure, but it feels to me like the research is reasonably novel. They reuse the 3rd-party CLIP system for the visual grounding, but CRAFT's use of a graph derived from ConceptNet, its path-based scoring scheme for measuring the confidence of verb-to-object mapping, its energy function, and its iterative refinement scheme feel original.

**impact / significance**

The impact / significance of the research is modest. The predictive performance of CRAFT is roughly the same as the other non-Oracle affordance grounding systems.  The advantages of CRAFT's approach to affordance grounding are not readily apparent.

**other observations**

The authors declare repeatedly, and in various ways, that CRAFT reasons. I'm not convinced of this and believe that this claim mischaracterises CRAFT. For example, the authors describe the multi-hop paths they walk in their knowledge graph of ConceptNet (where they walk from a target verb outward to find related affordance objects) as 'reasoning traces' and 'multi-hop reasoning'.  They refer to 'reasoning-based prior scores' and 'affordance reasoning'.  I don't see why walking paths in a graph warrants the label 'reasoning'. This is traversing and leveraging symbolic, common sense, background (prior) knowledge, yes; but it's not reasoning in the sense normally understood. Consider justifying the claims and language of reasoning, or dropping them.

The authors refer to their overall scoring function as an 'energy' function, and to their affordance grounding strategy as 'energy-based'. They refer to 'energy minimisation'.  Perhaps there are conventions of which I am unaware which normalise such language.  I kept hoping for an explanation of the 'energy' label the authors assign to their function, but I didn't notice one.

The metric/function used for computing the confidence score for the path-based verb-to-object mappings is described a bit in natural language but remains rather opaque.

**Anonymity:**

Remain anonymous

---

### Official Review · Reviewer_5Vkx · 2025-07-08

**Rating:** 3
**Confidence:** 4

**Review:**

## Strength
1. This paper is well-organised. The writing is clear and easy to understand.

## Weakness
1. Novelty: In the formalization, the problem facing CRAFT is cross-modal retrieval. CRAFT provides a solution by constructing an affordance graph and designing an algorithm for path scores and their updating. However, the idea of using LLMs to build graphs has been proposed in many papers, and retrieving vectors based on similarity or scores in semantic space is not a fresh idea.
2. Experiments: Experiments on only one dataset are insufficient to demonstrate the validity of CRAFT.

### questions
1. I think the title does not match the content. The method presented in this paper is a cross-modal retrieval algorithm, rather than a neural-symbolic reasoning framework. Constructing a graph by querying ConceptNet or LLMs does not qualify as symbolic reasoning. Symbolic reasoning builds on rules and regularities, and it has two prominent frameworks, logical reasoning in formal systems and probabilistic reasoning such as Bayesian networks and Markov random fields.
2. The paper does not show that CRAFT addresses the limitations described in the introduction, such as generalization to open-world tasks. Please specify which limitations CRAFT addresses.
3. I suggest you experiment on multiple datasets to strength the argument.
4. In the experiments, ResNet-RNN outperforms your method on the evaluation metrics. Could you point out where your method is superior to ResNet-RNN?

**Anonymity:**

Remain anonymous

---

### Official Review · Reviewer_hxmx · 2025-07-08
**Well written paper for a promising "label free" approach.**

**Rating:** 7
**Confidence:** 3

**Review:**

#Summary

The paper introduces a NeSy approach for affordance grounding: in practice this
amounts to retriving images depicting objects (or more generally visual
concepts) that can be used to implement a target action/verb. The approach
integrates prior knowledge (to map actions to potential objects) and
pre-trained embeddings (of object descriptions and image contents) to define an
objective useful for identiying good candidate images. The objective is the
product of an action-object potential and an object-image potential obtained
from different sources (ConceptNet and CLIP, respectively). Furthermore, the
first potential is iteratively refined based on visual evidence (i.e. strength
of image matches as suggested by CLIP). Importantly, the proposed approach
is "label free".

**TL;DR**: I did not find major issues with the paper.


#Clarity

Nicely written, well structured, quite clear.


#Significance

Definitely on-topic for the conference.  The target task is outside of my
comfort zone, and I cannot estimate precisely how significant it is for
research on affordances.


#Originality

The idea of using pre-trained text and image embeddings to map images to
objects/concepts is not new (see the literature on label-free concept
bottleneck models). The key novelty seems to lie in combining the two
aforementioned potentials together.

I am not familiar enough with the affordance literature to really estimate how
novel the contribution as a whole is in this context.


#Quality

The proposed approach is intuitively sensible.

The competitors were chosen appropriately (to the best of my understanding) and
are quite varied; the target dataset is also good.

Performance seems promising, although it lags behind a fully supervised
alternative (ResNet-RNN).  Integrating the proposed approach with LLMs also
seems to provide a boost (which I suspect is a good thing).

**Anonymity:**

Remain anonymous